# Pediatric Pancreatic Endocrine Tumor Presenting as Acute Pancreatitis: A Case Report

**DOI:** 10.3390/children10050900

**Published:** 2023-05-19

**Authors:** Shigetaka Fukuda, Mitsuyoshi Suzuki, Kei Minowa, Hiroyuki Koga, Atsuyuki Yamataka, Toshiaki Shimizu

**Affiliations:** 1Department of Pediatrics, Faculty of Medicine, Juntendo University, Tokyo 113-8421, Japan; s.fukuda.zw@juntendo.ac.jp (S.F.); kminowa@juntendo.ac.jp (K.M.); tshimizu@juntendo.ac.jp (T.S.); 2Department of Pediatric General and Urogenital Surgery, Faculty of Medicine, Juntendo University, Tokyo 113-8421, Japan; h-koga@juntendo.ac.jp (H.K.); yama@juntendo.ac.jp (A.Y.)

**Keywords:** pancreatic neuroendocrine tumor, acute pancreatitis, children

## Abstract

Pancreatic neuroendocrine tumors (PNETs) are relatively rare, especially in the pediatric age group. This report describes a pediatric case of acute pancreatitis secondary to stenosis of the main pancreatic duct due to a PNET. The patient was a boy, thirteen and a half years old, who presented with persistent low-grade fever, nausea, and abdominal pain. He was diagnosed with acute pancreatitis based on the elevation of serum pancreatic enzyme levels and abdominal ultrasonography findings of enlargement of the pancreas and dilatation of the main pancreatic duct. Abdominal contrast-enhanced computed tomography (CT) showed a 5.5 mm, contrast-enhanced mass in the head of the pancreas. His symptoms resolved with conservative treatment, although the pancreatic tumor grew slowly. At fifteen years and four months, since the tumor had enlarged to 8.0 mm, the patient underwent pancreaticoduodenectomy for therapeutic and diagnostic purposes. Based on the pathological evaluation, he was diagnosed with PNET (grade: G1). The patient has been free of tumor recurrence for 10 years and requires no additional therapy. In this report, the clinical characteristics of PNETs are also discussed, comparing the clinical features of adult-onset and pediatric-onset cases that initially present as acute pancreatitis.

## 1. Introduction

The causes of childhood acute pancreatitis are diverse, including anatomical abnormalities, trauma, cholelithiasis, drugs, and hereditary factors [1]. Determining the etiology of recurrent acute pancreatitis is crucial to prevent recurrent attacks and complications. Tumors, such as solid pseudopapillary tumors and lymphomas, are also known to contribute to the development of pancreatitis through the induction of pancreatic duct obstruction [1]. Pancreatic neuroendocrine tumors (PNETs) represent less than 3% of pancreatic neoplasms, with reports increasing due to recent advances in diagnostic imaging [2]. Patients with these tumors complained of various clinical symptoms, such as abdominal discomfort, pain, nausea, weight loss, and diarrhea [3,4]. These tumors generally develop in the pancreatic parenchyma, away from the main pancreatic duct; the occurrence of acute pancreatitis in cases of PNET is rare in adults [5], and only two pediatric patients were previously described [6,7]. In this report, a pediatric case of acute pancreatitis secondary to stenosis of the main pancreatic duct due to a tumor pathologically proven to be a serotonin-producing PNET is discussed.

## 2. Case Report

### 2.1. Patient Information

The patient was a boy, thirteen and a half years old, with persistent low-grade fever, nausea, and abdominal pain. The patient’s past and family histories were unremarkable.

### 2.2. Clinical Course before Surgery

The diagnosis of acute pancreatitis was made based on elevation of serum pancreatic enzyme levels (amylase: 541 U/L, and lipase: 1403 U/L) and abdominal ultrasonography findings of enlargement of the pancreas and dilatation of the main pancreatic duct. Abdominal contrast-enhanced computed tomography (CT) showed a 5.5 mm, contrast-enhanced mass protruding into the main pancreatic duct in the head of the pancreas (Figure 1A). He was hospitalized for two weeks and treated with fasting and intravenous fluids. His symptoms resolved with this conservative treatment, but the pancreatic tumor grew slowly during outpatient follow-up. After recovering from the initial onset of pancreatitis, magnetic resonance cholangiopancreatography (MRCP) showed a defect of the pancreatic duct in the head of the pancreas (Figure 2). Endoscopic ultrasound was performed to clarify the relationship between the tumor and the pancreatic duct. The pancreatic duct stenosis in the head of the pancreas and duct dilatation from the body to the tail of the pancreas were observed, but no mass lesion was identified despite the use of the contrast medium Sonazoid.

At fifteen years and four months of age, since the tumor had enlarged to 8.0 mm (Figure 1B), the patient underwent pancreaticoduodenectomy for therapeutic and diagnostic purposes. Before the operation, the oral glucose tolerance test showed glucose levels in the normal range, and the value of the oral N-benzoyl-L-tyrosyl-p-aminobenzoic acid test for measuring pancreatic exocrine function was 63.5% (normal range: 73.4–90.4%). His pancreatic tumor marker (Span-1: less than 1.0 U/mL, DUPAN-2: less than 25 U/mL, and CA19-9: 5 U/mL) and pancreatic endocrine hormone (insulin: 10.2 µU/mL, and gastrin: 79 pg/mL) levels were within normal limits. Neither metastatic lesions nor tumors suggesting multiple endocrine neoplasia type 1 were observed on CT and magnetic resonance imaging (MRI).

### 2.3. Surgical Treatment

The patient underwent laparotomy via a reverse L-shaped incision. The posterior peritoneum was opened adjacent to the descending part of the duodenum. Kocher’s mobilization was performed. The gallbladder artery, right gastric artery/vein, and gastroduodenal artery were ligated and divided. The right gastroepiploic artery/vein and accessory right colic vein were ligated and divided, while the middle colic vein was preserved. The stomach was divided by a linear stapler at a site 1–1.5 cm proximal to the pylorus. The pancreas was transected with an electric knife at the left margin of the superior mesenteric vein, and a 12-Fr pancreatic tube was inserted into the cut end of the main pancreatic duct. The jejunum was detached from the ligament of Treitz down to about 20 cm distal to the ligament, and the small intestine was divided with a linear stapler. The lower pancreatic edge was detached from the right edge of the portal vein to the head of the pancreas, and the common bile duct was divided. The distal end of the jejunum was lifted to the retrocolic route, and a pancreatojejunostomy was performed about 14 cm distal to the jejunal cut end. A small hole was made in the jejunum, and a pancreatic tube was inserted. The tube was brought out through the abdominal wall and fixed in the Witzel fashion. The pancreatojejunostomy was then completed with full-thickness sutures. A drain was placed beyond the jejunal cut end of the choledochojejunostomy. An end-to-end anastomosis was performed about 5 cm distal to the pancreatojejunostomy. A gastrojejunostomy was performed using the Billroth II technique, and a 1 cm Braun anastomosis was placed about 10 cm distal to the gastrojejunostomy. The omentum was placed between the gastroduodenal artery root and the pancreatojejunostomy, and the abdomen was closed. The patient started drinking water from postoperative day (POD) 5, initiated a liquid diet on POD7, and advanced to a regular diet on POD15. All drains were removed on POD19, and the enterostomy tube was clamped on POD22. The patient was discharged on POD29. The enterostomy tube and pancreatic tube were removed on POD56 at the outpatient clinic.

### 2.4. Histological Examinations

On postoperative examination, the resected specimen was found to be a well-defined mass (Figure 3). The tumor was located near the main pancreatic duct. However, there was no evidence of the tumor invading or destroying the duct. Thus, it was determined that the tumor was compressing the pancreatic duct, which led to dilation of the distal pancreatic duct. The pathological evaluation showed a well-differentiated endocrine neoplasm with marked stromal fibrosis, and the expression level of the cell proliferation marker Ki-67 was less than 2%. Immunohistochemical staining was positive for chromogranin A, which localizes to secretory vesicles in neurons and endocrine cells, and it was negative for synaptophysin. In addition, immunohistochemical staining was negative for glucagon, insulin, and somatostatin and positive for serotonin (Figure 4). Based on these findings, the diagnosis of a serotonin-producing PNET (grade G1) was made, and no additional treatment was given.

### 2.5. Follow-Up and Outcome

Postoperatively, the patient stopped taking oral proton pump inhibitors and pancreatic enzymes for a period, but resumed them later, due to complaints of loose stools. Contrast-enhanced CT of the abdomen 10 months after surgery showed persistent dilation of the ducts (Figure 5A); however, that of the abdomen performed 8 years later showed pancreatic atrophy (Figure 5B). He has been using proton pump inhibitors and pancreatic enzymes and has been well-managed for the past 10 years without any tumor recurrence or malnutrition, as confirmed by annual check-ups.

## 3. Literature Review

To understand the characteristics of PNET, which presents as acute pancreatitis among various clinical manifestations, the PubMed and Ichushi web databases were searched from inception until February 2023 using the keywords “pancreatic neuroendocrine tumor” and “acute pancreatitis”. In total, 763 related articles were retrieved. There were six articles [for 44 adult [5,6,8,9,10] and three pediatric [6,7] (age < 18 years) cases, including the present case] that reported the course of each case in detail (Table 1). There were 21 male and 21 female patients (sex was not reported for 5 patients), with onset ages of 11 to 76 years. In patients over 20 years of age, the median age of onset was 48.0 years; 77.3% had non-functioning PNETs, 29.5% had involvement of the head of the pancreas, and 13.6% had severe acute pancreatitis [5,6,8,9,10].

So far, two Japanese pediatric patients have been reported, excluding the present patient [6,7]. One patient was an 11-year-old girl whose initial symptoms were abdominal pain, vomiting, and fever, and she developed severe acute pancreatitis. An approximately 5 cm mass was found in the body of the pancreas, and a distal pancreatectomy was performed for diagnostic and therapeutic purposes. Pathologically diagnosed as a benign non-functioning tumor, the patient is alive ten years after surgery with no complications, such as diabetes mellitus [6]. The other was a 16-year-old boy who complained of abdominal pain and was diagnosed with acute pancreatitis. Abdominal CT showed a tumor, 8 cm in diameter, in the head of the pancreas. Robot-assisted pancreaticoduodenectomy was performed after the pancreatitis healed. Histologically, a grade G2 non-functional PNET was confirmed [7].

## 4. Discussion

The pathogenesis of acute pancreatitis caused by pancreatic tumors includes (1) mechanical obstruction of the pancreatic duct, (2) ischemia associated with vascular obstruction caused by cancer, and (3) pancreatic enzyme activation by tumor cells [11]. Serotonin-producing tumors are characterized by tumor growth with fibrous stroma around the pancreatic duct and pancreatic duct stenosis due to compression of the pancreatic duct [12]. The production of several substances, including serotonin, 5-hydroxyindolacetic acid, and further downstream signaling elements, is thought to be involved in stromal fibrosis [12]. 

Imaging characteristics of PNETs include well-defined borders and dense staining in the early phases due to the rich blood flow on contrast-enhanced CT and MRI [13]. Ultrasonography and endoscopic ultrasonography show a round or oval hypoechoic mass with smooth margins, well-defined borders, and a heterogeneous interior [13]. In the present case, the tumor seemed to infiltrate into the main pancreatic duct on CT at the time of the initial pancreatitis attack. At the same time, the gross examination did not show any prominent invasive findings, and the boundary between the tumor and the normal tissue was clearly separate on pathological examination. However, the area surrounding the tumor showed obstructive pancreatitis with atrophy and fibrosis of the pancreas acinus, suggesting pancreatic duct compression by the tumor.

Generally, partial pancreatectomy is recommended for patients diagnosed with PNET without distant metastases [14]. However, in small non-functional PNETs localized to the head of the pancreas, invasive pancreaticoduodenectomy tends to be avoided first, and careful follow-up is conducted. If the tumor is non-functioning and there are no indications of metastasis or invasion on imaging, patients with tumors less than 1 cm in size, asymptomatic, and discovered incidentally are not subjected to immediate surgery [15,16]. Follow-up every 6–12 months is also an option, and surgery is considered when the tumor grows, or clinical symptoms develop [15,16].

Of PNETs, serotonin-positive tumors are characteristically known to cause pancreatic ductal stenosis, which leads to duct dilation and atrophy of the upstream pancreas independent of tumor size [4]. In a previous study of six PNET patients in whom serotonin was expressed, all cases showed significant stromal fibrosis in the tumor area, making them stand out. This fibrosis caused a concentric narrowing of the pancreatic duct. Small PNETs can lead to pancreatic duct obstruction, causing upstream duct dilation and pancreatic atrophy out of proportion to the size of the tumor [4]. In the present case, since the pancreatic tumor grew and the pancreatic duct dilation persisted, an elective procedure was chosen to avoid further pancreatic morphological change and preserve function. Consequently, mild pancreatic exocrine dysfunction remained, but the surgery may have contributed to maintaining the patient’s quality of life, since the patient did not develop diabetes mellitus. Furthermore, prophylactic surgery would be meaningful in severe acute pancreatitis, a potentially lethal disease that occurs in 13.6% of adults and 33.3% of children presenting with severe acute pancreatitis.

## 5. Conclusions

Acute pancreatitis associated with PNETs is rare, especially in children, but it should be included in the differential diagnosis of acute pancreatitis. Although most of these tumors are adjacent to the pancreatic duct without invasion, complete surgical resection should be attempted due to the possibility of malignancy, atrophy of the pancreas, and life-threatening severe acute pancreatitis.

## Figures and Tables

**Figure 1 children-10-00900-f001:**
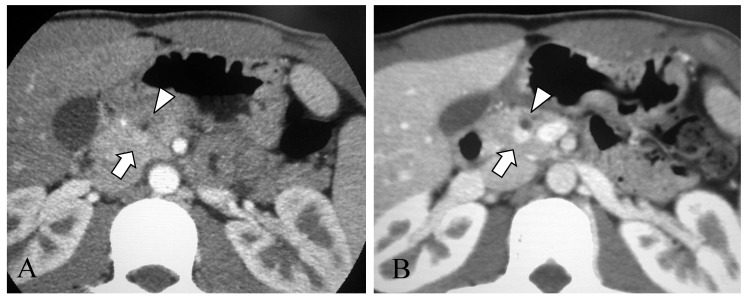
(**A**) Contrast-enhanced computed tomography at 13 years 6 months of age. A 5.5-mm, contrast-enhanced mass (arrow) is seen in the head of the pancreas. The upstream main pancreatic duct (arrowhead) is dilated. No metastatic lesions are evident. (**B**) Contrast-enhanced computed tomography at 15 years 4 months of age. The mass has enlarged to 8.0 mm (arrow), and dilatation of the main pancreatic duct continues to be seen. No metastatic lesions are visible.

**Figure 2 children-10-00900-f002:**
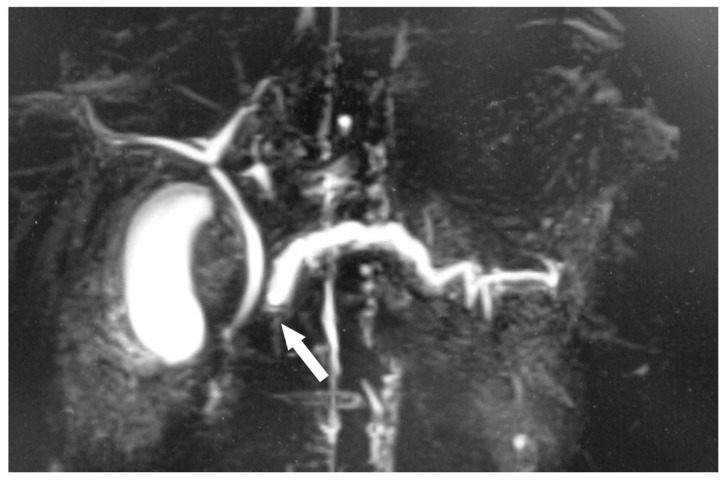
Magnetic resonance cholangiopancreatography at 13 years 6 months of age. Severe stenosis of the main pancreatic duct and dilatation of the distal pancreatic duct are seen.

**Figure 3 children-10-00900-f003:**
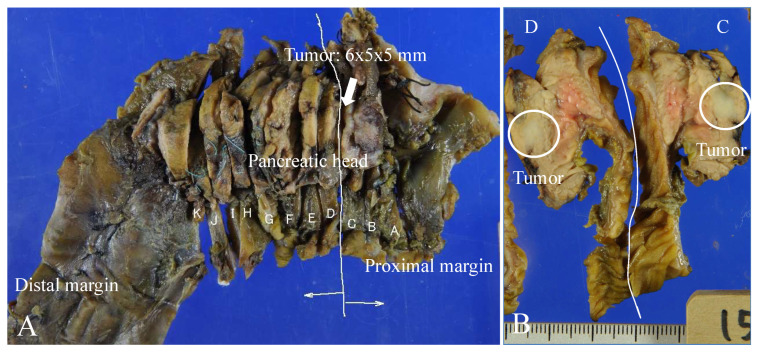
(**A**) Formalin-fixed excised specimen of the head of the pancreas and duodenum. The marks A to K in the figure indicate the cutting position of the specimen. (**B**) Photograph of the gross resected specimen showing between cutting positions C and D. The cross-section shows a 6 × 5 × 5 mm^3^ solid tumor (circle).

**Figure 4 children-10-00900-f004:**
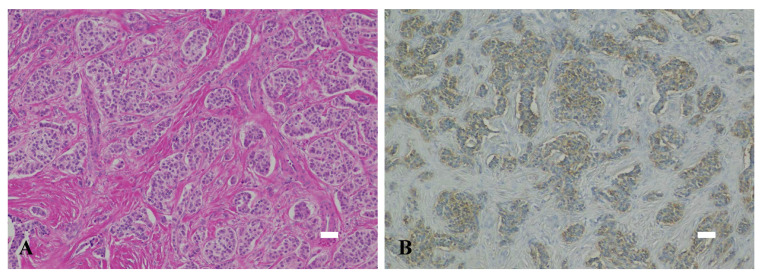
(**A**) Vesicular growth of tumor cells with round nuclei and eosinophilic cytoplasm (hematoxylin-eosin stain). (**B**) The immunohistochemical staining is positive for serotonin. The length of the bar is 100 μm.

**Figure 5 children-10-00900-f005:**
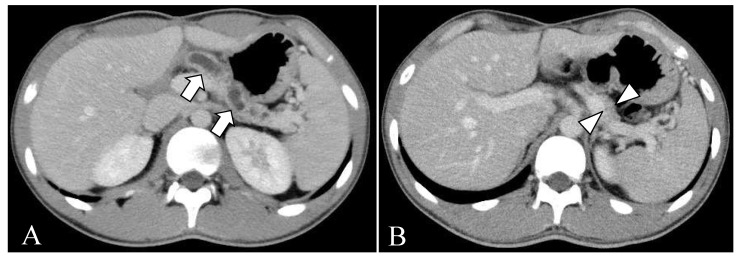
(**A**) The contrast-enhanced CT image shows dilatation of the pancreatic duct (arrow) and pancreatic atrophy 9 months after the operation. (**B**) An atrophic truncated appearance of the body and tail of the pancreas (arrowheads) without dilatation of pancreatic duct 8 years after surgery.

**Table 1 children-10-00900-t001:** Characteristics of acute pancreatitis secondary to PNETs.

	Adults (*n* = 44)		Children (*n* = 3)
	Median	Range	
Age (years)	48	20–76	11, 13, 16
	*n*	%	*n*
Female	20 *	45.4	1
Location of tumor			
Head	13	29.5	2
Body	9	20.5	1
Tail	12	27.3	0
Others	4	9.1	0
Not available	6	13.6	0
Pathological diagnosis			
Non-functioning	34	77.3	3
Gastrinoma	1	2.3	0
Glucagonoma	1	2.3	0
Somatostatinoma	1	2.3	0
Not available	7	15.9	0
Pancreatitis severity			
Severe	6	13.6	1
Not severe	35	79.5	2
Not available	3	6.8	0

* Sex was not available for 5 patients.

## Data Availability

The data presented in this study are available on request from the corresponding author. The data are not publicly available due to privacy.

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
