# Peer review of "Pediatric Pancreatic Endocrine Tumor Presenting as Acute Pancreatitis: A Case Report"

_children, 2023, doi:10.3390/children10050900_

Round 1

Reviewer 1 Report

I read this paper with much interest. It was well-written and informative.

some minor revision are required.

1. Please add the right scale bar for the HE and IHC staining in the Figure 4 

Author Response

Reviewer 1

I read this paper with much interest. It was well-written and informative.

  1. Please add the right scale bar for the HE and IHC staining in the Figure 4.

Response: Figure 4 now includes the appropriate scale bar.

Reviewer 2 Report

Fukuda et al. presented a rare case of acute pancreatitis caused by the obstruction of pancreatic duct by neuroendocrine tumor. I have a few comments as the below.

1. The most concern in the present case would be how the pancreatic duct was obstructed. Please describe the detail of histologic findings on the relationship of the tumor with the occluded main pancreatic duct. Did the tumor just compressed the pancreatic duct?

2. Please also show the long-term outcomes of the patient. Did the dilation of main pancreatic duct improve postoperatively? I would also like to know whether the patient is still taking oral pancreatic enzymes. 

Author Response

Reviewer 2

Fukuda et al. presented a rare case of acute pancreatitis caused by the obstruction of pancreatic duct by neuroendocrine tumor. I have a few comments as the below.

  1. The most concern in the present case would be how the pancreatic duct was obstructed. Please describe the detail of histologic findings on the relationship of the tumor with the occluded main pancreatic duct. Did the tumor just compressed the pancreatic duct?

Response: On histological examination, the tumor in this case was located near the main pancreatic duct. However, there was no evidence of the tumor invading or destroying the duct. Thus, it was determined that the tumor was compressing the pancreatic duct, which led to dilation of the distal pancreatic duct. These sentences were inserted in ‘2.4. Histological examinations’ section.

  1. Please also show the long-term outcomes of the patient. Did the dilation of main pancreatic duct improve postoperatively? I would also like to know whether the patient is still taking oral pancreatic enzymes. 

Response: Nine months after the operation, contrast-enhanced CT showed that the main pancreatic duct had dilated (as seen in Figure 5A). Eight years post-surgery, CT showed an atrophic truncated appearance of the body and tail of the pancreas. The patient had stopped taking oral pancreatic enzymes for a period but resumed them later due to complaints of loose stools. Currently, the patient is still taking the enzymes.

Reviewer 3 Report

Thank you for the opportunity to review the manuscript entitled, "Pediatric pancreatic endocrine tumor presenting as acute pancreatitis: A case report and review of the literature" by Dr. Fukuda and colleagues.  In the manuscript, the authors present the clinical presentation and treatment course for a 13 year old boy with a G1 pancreatic head PNET presenting initially with acute pancreatitis.  This represents only the 3rd reported pediatric case of this condition.  

The manuscript is well written and generally follows the elements of the CARE reporting guidelines for case reports.  The figures, tables and references are well selected and constructed.

Please address the following comments:

1)  Did you obtain informed consent for this report?  Just an inquiry as some authors routinely do this.

2)  I encourage the authors to expand the discussion regarding the treatment course for the patient.  The authors observed the 5 cm tumor for over 1.5 years before removing.  I assume the patient was asymptomatic in the time interval between the initial pancreatitis and the resection.  Where their interval imaging studies between presentation and resection?  Would the authors do anything differently the next time they encounter a similar patient?

3)  Lines 136-138.  The sentence is somewhat unclear.  Are the authors stating that asymptomatic, non-functioning tumors with no metastatic or invasive findings undergo up-front resection?  It is the word "unless" that confuses me.  Please clarify as I feel that the authors are trying to describe the validity of an initial observation approach for these patients.

Thank you for allowing me to participate in this review.

Author Response

Reviewer 3

Thank you for the opportunity to review the manuscript entitled, "Pediatric pancreatic endocrine tumor presenting as acute pancreatitis: A case report and review of the literature" by Dr. Fukuda and colleagues.  In the manuscript, the authors present the clinical presentation and treatment course for a 13 year old boy with a G1 pancreatic head PNET presenting initially with acute pancreatitis.  This represents only the 3rd reported pediatric case of this condition.

  1. Did you obtain informed consent for this report?  Just an inquiry as some authors routinely do this.

Response: We did obtain written, informed consent from this patient and have also included the Informed Consent statement in its default position within the text.

  1. I encourage the authors to expand the discussion regarding the treatment course for the patient. The authors observed the 5 cm tumor for over 1.5 years before removing. I assume the patient was asymptomatic in the time interval between the initial pancreatitis and the resection. Where their interval imaging studies between presentation and resection? Would the authors do anything differently the next time they encounter a similar patient?

Response: The tumor size before the operation was 8.0 mm. Please see the ‘2.2. Clinical course before surgerysection and ‘Figure 1’. Between initial pancreatitis and the resection, we regularly performed echotomography every 2-3 months at an outpatient clinic. Although we selected surgery for treatment, his remaining pancreas atrophied, and he required pancreatic enzyme therapy. If we encounter a patient with similar symptoms, we will use the same approach.

  1. Lines 136-138. The sentence is somewhat unclear. Are the authors stating that asymptomatic, non-functioning tumors with no metastatic or invasive findings undergo up-front resection? It is the word "unless" that confuses me. Please clarify as I feel that the authors are trying to describe the validity of an initial observation approach for these patients.

Response: We revised the sentence ‘Patients with tumors less than 1 cm, asymptomatic, and incidentally discovered would not be operated on immediately unless the tumor is non-functioning with no metastatic or invasive findings on imaging’ to ‘If the tumor is non-functioning, and there are no indications of metastasis or invasion on imaging, patients with tumors that are less than 1 cm in size, asymptomatic, and discovered incidentally are not subjected to immediate surgery’.

Reviewer 4 Report

Thank you for the opportunity to review this case report. The case is rare, and the authors highlight the fact that acute pancreatitis in children may present as the first symptom of a pancreatic tumor. There are good quality figures to accompany the text. I find, however, that the report lacks important information on the diagnostic and therapeutic process. Below are more detailed comments:

 Abstract :He has been successfully managed with this regimen” – not clear, what kind of regimen

Case report:

What was the conservative treatment and how long was it administered, was the patient hospitalized, and if yes, for how long?

As the authors state in the discussion, small, non-symptomatic pNETs may be left for surveillance. However, in this patient the tumor was symptomatic from the first diagnosis (pancreatitis) and it seems that it was not established that the tumor is a pNET until the surgery was performed. Please discuss, why  Ca 19-9, CEA, Chromogranin A in serum were not tested in this patient and EUS was not performed prior to the very risky surgery, to try and diagnose the nature of the tumor preoperatively (The authors state the surgery was due to “therapeutic and diagnostic purposes”). What was the surveillance after the initial acute pancreatitis treatment, before the decision on pancreatoduodenectomy was made (CT scans, ultrasound, tumor markers)?

Was the patient administered proton pump inhibitors and pancreatic enzyme supplementation for 10 years postoperatively and if yes, why? Did he experience steatorrhea postoperatively? Was the postoperative course uneventful?  What was the annual check-up, did the patient have CT-scans or abdominal ultrasound, or tumor markers tested?

Author Response

Reviewer 4

Thank you for the opportunity to review this case report. The case is rare, and the authors highlight the fact that acute pancreatitis in children may present as the first symptom of a pancreatic tumor. There are good quality figures to accompany the text. I find, however, that the report lacks important information on the diagnostic and therapeutic process. Below are more detailed comments:

  1. Abstract: ‘He has been successfully managed with this regimen’ – not clear, what kind of regimen.

Response: We have omitted the sentence ‘He has been successfully managed with this regimen’ in the abstract because it was unclear.

  1. What was the conservative treatment and how long was it administered, was the patient hospitalized, and if yes, for how long?

Response: The patient was hospitalized for two weeks and treated with fasting and intravenous fluids. We have added this information in the ‘2.2. Clinical course before surgery’ section.

  1. As the authors state in the discussion, small, non-symptomatic pNETs may be left for surveillance. However, in this patient the tumor was symptomatic from the first diagnosis (pancreatitis) and it seems that it was not established that the tumor is a pNET until the surgery was performed. Please discuss, why  Ca 19-9, CEA, Chromogranin A in serum were not tested in this patient and EUS was not performed prior to the very risky surgery, to try and diagnose the nature of the tumor preoperatively (The authors state the surgery was due to “therapeutic and diagnostic purposes”). What was the surveillance after the initial acute pancreatitis treatment, before the decision on pancreatoduodenectomy was made (CT scans, ultrasound, tumor markers)?

Response: The blood chromogranin A assay has proven to be helpful, but it is unfortunate that it is not covered by insurance in Japan. During the regular check-up, CA19-9 was tested, but CEA was not included in the routine testing. Endoscopic ultrasound was performed before surgery; however, little information was available on the relationship between the tumor and the pancreatic duct, and on the presence of a tumor. Between the initial pancreatitis and the resection, we regularly performed ultrasonography every 2-3 months at an outpatient clinic. Based on your suggestion, we have appropriately addressed these concerns within the text.

  1. Was the patient administered proton pump inhibitors and pancreatic enzyme supplementation for 10 years postoperatively and if yes, why? Did he experience steatorrhea postoperatively? Was the postoperative course uneventful? What was the annual check-up, did the patient have CT-scans or abdominal ultrasound, or tumor markers tested?

Response: We have addressed your question by thoroughly revising section ‘2.5. Follow-up and outcome’, which covers the patient’s follow-up and outcome in detail. An annual check-up is scheduled to test tumor markers, ultrasonography at every visit, and a CT or MRI scan each year.

Reviewer 5 Report

Thank you for submission of this interesting case report to Children. The case report describes a very rare PNET of the pancreas in a child. The article is interesting and well presented.

I have only one suggestion:

Please delete the last column of Table 1 where you present percentages for 3 children. It is unusual to present percentages for numbers <10.

Author Response

Reviewer 5

Thank you for submission of this interesting case report to Children. The case report describes a very rare PNET of the pancreas in a child. The article is interesting and well presented.

I have only one suggestion: Please delete the last column of Table 1 where you present percentages for 3 children. It is unusual to present percentages for numbers <10.

Response: We deleted the last column of Table 1.

Reviewer 6 Report

Dear authors,

This is a nice case report and due to the rarity of the case in pediatric population, I would recommend publishing the manuscript, but only after a major revision. My recommendations are as follows:

Title - should only specify the case report. A  brief review of the literature is implicitly discussed in the Discussions chapter.

Abstract - it should begin with an introductory phrase that frames the reported case.

Case report - more details regarding the operative indication should be presented. Was the patient operated only because the tumor grew from 5.5 mm to 8 mm, or were there other symptoms? What about the biological markers other than pancreatic tumor markers? Also, details regarding the surgical approach, the intraoperative aspect of the pancreas and the lesion, the operative technique as well as the immediate postoperative evolution should be presented.

Discussion - more details about pancreatitis and PNET pathology should be presented, and also the discussion on other similar cases should be more detailed.  

The approval of the ethics committee of the hospital for the publication of this case is missing.

Author Response

Reviewer 6

This is a nice case report and due to the rarity of the case in pediatric population, I would recommend publishing the manuscript, but only after a major revision. My recommendations are as follows:

  1. Title - should only specify the case report. A brief review of the literature is implicitly discussed in the Discussions chapter.

Response: We have made the necessary correction to the title.

  1. Abstract - it should begin with an introductory phrase that frames the reported case.

Response: We added a short introduction at the beginning of the Abstract.

  1. Case report - more details regarding the operative indication should be presented. Was the patient operated only because the tumor grew from 5.5 mm to 8 mm, or were there other symptoms? What about the biological markers other than pancreatic tumor markers? Also, details regarding the surgical approach, the intraoperative aspect of the pancreas and the lesion, the operative technique as well as the immediate postoperative evolution should be presented.

Response: The patient had no clinical symptoms during outpatient follow-up until the operation. Generally, most of these tumors are adjacent to the pancreatic duct and are not invasive, but the potential for malignancy, pancreatic atrophy, and life-threatening severe acute pancreatitis necessitates an attempt at complete surgical resection. The reasons for the surgery based on these factors were stated during the Discussion. The blood chromogranin A assay has proven to be helpful, but it is unfortunate that it is not covered by insurance in Japan. Our patient underwent pancreaticoduodenectomy to safeguard both pancreatic exocrine and endocrine functions, as well as for therapeutic and diagnostic purposes. Detailed information about the surgical procedure and its postoperative course can be found under the heading '2.3. Surgery treatment'.

  1. Discussion - more details about pancreatitis and PNET pathology should be presented, and also the discussion on other similar cases should be more detailed.

Response: On histological examination, the tumor in this case was located near the main pancreatic duct. However, there was no evidence of the tumor invading or destroying the duct. Thus, it was determined that the tumor was compressing the pancreatic duct, which led to dilation of the distal pancreatic duct. These sentences were inserted in the ‘2.4. Histological examinations’ section. In the '3. Literature review' section, we provided a detailed description of other cases of pediatric PNETs that were discovered due to acute pancreatitis.

  1. The approval of the ethics committee of the hospital for the publication of this case is missing.

Response: The ethics committee of our hospital approved the publication of this case. We also obtained written, informed consent from this patient and have also included the Informed Consent statement in its default position within the text.

Round 2

Reviewer 2 Report

The authors adequately responded to each reviewer's comments. I have no addtional comments for further revision and I think the paper can be accepted in the present form.

Reviewer 4 Report

Dear Authors,

Thank you for reviewing the manuscript. I find the study appropriate for publication.

Reviewer 6 Report

Dear authors,

Thank you for your response. I think the manuscript is now ready for publication.